# The Impact of Antibiotic Prophylaxis on a Retrospective Cohort of Hospitalized Patients with COVID-19 Treated with a Combination of Steroids and Tocilizumab

**DOI:** 10.3390/antibiotics12101515

**Published:** 2023-10-06

**Authors:** Francisco Javier Membrillo de Novales, Germán Ramírez-Olivencia, Maj. Tatiana Mata Forte, María Isabel Zamora Cintas, Maj. María Simón Sacristán, María Sánchez de Castro, Miriam Estébanez Muñoz

**Affiliations:** 1CBRN and Infectious Diseases Department, Hospital Central de la Defensa “Gómez Ulla”, 28047 Madrid, Spain; 2Clinical Microbiology Department, Hospital Central de la Defensa “Gómez Ulla”, 28047 Madrid, Spain; 3Pharmacy Department, Hospital Central de la Defensa “Gómez Ulla”, 28047 Madrid, Spain

**Keywords:** COVID-19, antimicrobial prophylaxis, superinfections, coinfections, ceftobiprole, ceftriaxone

## Abstract

Objectives: In the context of COVID-19, patients with a severe or critical illness may be more susceptible to developing secondary bacterial infections. This study aims to investigate the relationship between the use of prophylactic antibiotic therapy and the occurrence of bacterial or fungal isolates following the administration of tocilizumab in hospitalized COVID-19 patients who had previously received steroids during the first and second waves of the pandemic in Spain. Methods: This retrospective observational study included 70 patients hospitalized with COVID-19 who received tocilizumab and steroids between January and December 2020. Data on demographics, comorbidities, laboratory tests, microbiologic results, treatment, and outcomes were collected from electronic health records. The patients were divided into two groups based on the use of antibiotic prophylaxis, and the incidence of bacterial and fungal colonizations/infections was analyzed. Results: Among the included patients, 45 patients received antibiotic prophylaxis. No significant clinical differences were observed between the patients based on prophylaxis use regarding the number of clinically diagnosed infections, ICU admissions, or mortality rates. However, the patients who received antibiotic prophylaxis showed a higher incidence of colonization by multidrug-resistant bacteria compared to that of the subgroup that did not receive prophylaxis. The most commonly isolated microorganisms were *Candida albicans*, *Enterococcus faecalis*, *Staphylococcus aureus*, and *Staphylococcus epidermidis.* Conclusions: In this cohort of hospitalized COVID-19 patients treated with tocilizumab and steroids, the use of antibiotic prophylaxis did not reduce the incidence of secondary bacterial infections. However, it was associated with an increased incidence of colonization by multidrug-resistant bacteria.

## 1. Introduction

In the context of COVID-19, patients with a severe or critical illness may be more susceptible to developing secondary bacterial infections. Several meta-analysis studies have found a prevalence of bacterial coinfection in hospitalized patients ranging from 3.5% to 12% [1,2]. The risk of coinfection increases in patients requiring admission to critical care units, reaching up to 14–23% [2,3]. In most published observational studies and systematic reviews, the timing of bacterial infection diagnosis is not reported, so the term “coinfection” does not discriminate between community-acquired or nosocomial acquisition in these studies. However, in a study that took into account the time since admission [4], they found percentages of nosocomial bacterial superinfection that are similar to the rates published by studies that did not consider this factor.

The immunosuppressive treatment used for managing COVID-19 has been considered a potential risk factor for developing nosocomial acquired infections [5]. In the current management of patients admitted with COVID-19, the dexamethasone and tocilizumab guidelines are strongly recommended [6]. The association of tocilizumab treatment with the occurrence of bacterial infections was previously discovered [7]. The available data on the influence of tocilizumab on the risk of superinfection in COVID-19 to date are controversial. Stone et al. [8] did not find a higher incidence of superinfections in patients treated with tocilizumab, but in their series, most of these patients did not receive steroids. In the study by Ripa et al. [9], in the subgroup of patients who received immunosuppressive biological therapy, 14% presented at least one secondary infection compared to 9% of the overall population, with a median time from the start of therapy of 9 days; although in this cohort, only 22% of the included patients had received steroids. However, the meta-analysis by Tleyjeh IM. et al. [10] and the series by Narain et al. [11] associate the use of a combination of tocilizumab and steroids with a higher incidence of superinfections. In the meta-analysis published by Peng et al., a significant increase in fungal infections was noted after the use of tocilizumab [12], although it was not associated with an increase in bacterial infections.

Given that there are no consistent data to support an appropriate recommendation for the use of antibiotics in patients without a clinically suspected or documented infection [6], antibiotic prophylaxis is not routinely recommended in hospitalized patients with COVID-19. However, it could be considered in patients with risk factors for secondary bacterial infections.

In this study, we examined the relationship between the use of prophylactic antibiotic therapy and bacterial or fungal isolates following the administration of tocilizumab in a cohort of hospitalized COVID-19 patients who had previously received steroids during the first and second waves of the pandemic in Spain.

## 2. Methods

### 2.1. Study Design and Patients

This retrospective observational study was performed at the Hospital “Gómez Ulla” in Madrid (Spain), a 600-bed university center that provides broad and specialized medical, surgical, and intensive care for an urban population of 120,000 individuals. We included all patients who were 14 years or older, hospitalized in the conventional ward and/or intensive care unit (ICU) between 31 January and 6 December 2020 (during the first and second waves of the pandemic in Spain) with a clinical diagnosis of COVID-19 and confirmed via real-time reverse transcription PCR for SARS-CoV-2 using respiratory samples and who had received tocilizumab during hospitalization. The study was approved by the Ethics and Research Committee of the Study Hospital with code 25/20. The participants did not sign informed consent as it was a retrospective study, and there was no temporal overlap between their admission and data collection.

### 2.2. Data Collection and Outcomes

Using the off-guide drug dispensing software of the Hospital Pharmacy Service, the number of medical records of all the patients who had been treated with at least one 400 mg dose of tocilizumab during the study period was obtained.

For all the patients hospitalized with COVID-19 who met the inclusion criteria data concerning demographics (age and gender), epidemiology, comorbidities, laboratory tests, microbiologic results (blood and urine cultures, respiratory samples, urinary antigen tests, and antimicrobial susceptibility analyses), treatment and outcomes (intensive care unit admission, length of hospital stay, and mortality) were collected directly from electronic health records and drug dispensing software.

The records of all the patients with positive microbiologic results were reviewed by two clinicians with specific training in infectious diseases researchers (MEM and JMN) to assess the clinical significance. Microbiological isolates considered contaminants using microbiological or clinical criteria were excluded.

### 2.3. Procedures

The investigation of bacterial and fungal pathogens in blood, normally sterile fluids, sputum, and other samples, was performed with standard microbiologic procedures during hospitalization, as requested by the attending physician.

The samples were processed in the usual way in a laboratory following the procedures of the Spanish Society of Infectious Diseases and Clinical Microbiology (SEIMC). The samples were seeded on bacteriologic media, such as blood agar plate, chocolate agar plates, and MacConkey agar plates, using sterile wire loops and the filamentous fungi were incubated at 30 °C and the yeasts were incubated at 37 °C for 48 h in a thermostatic incubator. Routine fungus cultures were inoculated on Sabouraud/glucose (4%). The plates were incubated at 37 °C. Subsequently, the dominant and potentially pathogenic colonies were picked for bacterial and fungus detection using the VITEK MS system (bioMérieux, Marcy l’Étoile, France) or Microscan System (American MicroScan, Mahwah, NJ, USA). All these samples were processed according to the working procedures, and processed samples were published by the SEIMC. > n.d).

Microbiological isolates were considered within 14 days following the administration of tocilizumab [13]. Antibiotic prophylaxis was considered in patients who had received a prescribed antibiotic before or following tocilizumab infusion with the aim of using prophylaxis to prevent a further bacterial superinfection. Concomitant treatment with steroids was considered in patients who had received the tocilizumab infusion before, or at least one dose of dexamethasone equal to or greater than 6 mg/day or its equivalent. Coinfection was considered if the microbiological identification occurred within the first 48 h of admission. Superinfection was considered if the isolate corresponded to a sample obtained at least 48 h after hospital admission, and if, according to the clinical history data, there was a clinical suspicion of infection prior to the communication of the isolate, and/or the responsible clinician had prescribed targeted antibiotic therapy against the isolated microorganism. The appearance of multidrug-resistant (MDR) bacteria was analyzed: carbapenem-resistant *Acinetobacter baumannii* (CRAB), carbapenem-resistant *Enterobacteriaceae* (CRE), extended-spectrum beta-lactamase (ESBL)-producing *Enterobacteriaceae*, vancomycin-resistant *enterococci* (VRE), methicillin-resistant *Staphylococcus aureus* (MRSA), and carbapenem-resistant *Pseudomonas aeruginosa* (CRPA).

The Charlson comorbidity index [14] and the SEIMC Score [15] were calculated for all the patients included. The Charlson index is a system for assessing ten-year life expectancy, depending on the age at which it is evaluated and the comorbidities of the subject. The SEIMC Score is a prognostic scale developed by Sociedad Española de Enfermedades Infecciosas y Microbiología Clínica (Spanish Society of Infectious Diseases and Clinical Microbiology) that evaluates the risk of 30-day mortality based on parameters measured upon admission to the Emergency Department. The variables considered in this score are age, gender, the presence of dyspnea, baseline capillary oxygen saturation, the neutrophil-to-lymphocyte ratio, and creatinine clearance. An SEIMC Score of 9–30 points is associated with a very high 30-day mortality risk.

### 2.4. Statistical Analysis

The median and interquartile range were used as quantitative variables. Absolute frequencies and relative frequencies in percentages (%) were used as qualitative variables. Hypothesis testing was conducted using Fisher’s exact. A *p*-value of less than 0.05 was considered statistically significant. Statistical analysis was performed using SPSS^®^ version 25 software.

## 3. Results

During the study period, a total of 2069 COVID-19 patients were admitted to our hospital. Among these patients, 76 had a prescription for tocilizumab in the drug dispensation records of the Hospital Pharmacy Service. After reviewing the medical records, six patients who had not received the drug were excluded, resulting in a final study population of seventy patients (Figure 1).

### 3.1. Comorbidities and Risk at Admission

The 70 patients had a confirmed diagnosis of COVID-19 with a positive SARS-CoV-2 PCR test and were receiving steroid treatment at the time of tocilizumab administration. The median age was 66 years (IQR 54–77 years). The median Charlson index was 2 (IQR 2–5), and the median SEIMC Score was 10 (interquartile range 5–15). Twenty patients (28.6%) required admission to the intensive care unit, and thirty-two (45.7%) died during hospitalization. All patients received tocilizumab either in the Emergency Department or on the general hospital ward. A total of 20 patients (28.5%) were subsequently admitted to the intensive care unit. The median number of days from tocilizumab administration to microbiological isolation was eight.

Forty-five patients (64.3%) received antibiotic prophylaxis. Age and the SEIMC Score at admission were analyzed, and no statistically significant differences were found in relation to receiving antibiotic prophylaxis. However, there was a tendency for patients who received antibiotic prophylaxis to have a higher Charlson index (Charlson subgroup with prophylaxis: two; subgroup without prophylaxis: two, *p* = 0.09; Table 1).

A total of 24.4% of patients who received antibiotic prophylaxis ended up being admitted to the intensive care unit compared to 36% of patients who did not receive prophylaxis. However, no statistically significant differences were found. There were also no statistically significant differences in mortality between the two subgroups (Table 1).

### 3.2. Prophylaxis Used and Microbiological Results

Fourteen out of the seventy patients who received tocilizumab were identified to have bacterial and/or fungal infection/colonization within 14 days after tocilizumab administration. Thirteen out of the fourteen patients were admitted to the ICU after administration (patient 12 did not enter the ICU), hence the sample collection was conducted during their ICU stay; see Table 2. Out of the 45 patients who received prophylaxis, 10 patients (22%) had significant microbiological isolates interpreted as colonization or an infection. Out of the twenty-five patients who did not receive prophylaxis, four patients (16%) had significant microbiological isolates (Table 1). The most commonly used prophylactic antibiotics were ceftriaxone and ceftobiprole. The other antibiotics used alone or in combination included piperacillin plus tazobactam, teicoplanin, meropenem, linezolid, ciprofloxacin, amoxicillin plus clavulanic acid, and cefepime. The subgroups with the two most commonly used prophylaxis, ceftriaxone, and ceftobiprole treatments were analyzed independently (Table 3). No patient had a documented coinfection or superinfection prior to the administration of tocilizumab. 

The most frequently isolated microorganisms were *Candida albicans* (*n* = 7), *P. aeruginosa* (*n* = 3), *ESBL Escherichia coli* (*n* = 3), *Enterococcus faecalis* (*n* = 3), *Staphylococcus aureus* (*n* = 3), and *Staphylococcus epidermidis* (*n*= 2). Of the eighteen samples with bacterial isolations, four samples (22%) presented a multidrug-resistant pathogen (three *ESBL E. coli*; one *MRSA*). The four isolates with the growth of multidrug-resistant bacteria were interpreted as colonization. These four samples were identified from four patients who had received antibiotic prophylaxis. In three out of the four patients, samples were collected during their ICU stay (patients 1, 8, and 9). 

No statistically significant differences were found in terms of isolates, colonization, ICU admissions, or mortality among the different studied prophylaxis groups.

Below, we describe the details of the patients who had microbiological isolations.

Patient 1 was a 53-year-old male. Charlson index: three. SEIMC Score: five. They used prophylaxis with ceftriaxone, lasting for 6 days. They were admitted to the ICU 4 days after tocilizumab administration. During their stay in the ICU, bacteremia due to *S. epidermidis* was observed, and *C. albicans* was isolated in the bronchoaspirate and endotracheal aspirate samples. Additionally, positive IgM antibodies for *Chlamydia pneumoniae* were detected. The attending clinicians decided to treat the fungal isolation with intravenous anidulafungin, in addition to combined antibiotic coverage for the rest of the microbiological isolates. On day 12, after tocilizumab administration, *ESBL E. coli* was isolated in a rectal swab, which was interpreted as colonization. The patient died after 35 days of ICU admission.

Patient 2 was a 64-year-old male. Charlson index: two. SEIMC Score: six. He received prophylaxis with ceftriaxone for 6 days. The day after the administration of tocilizumab, he was admitted to the ICU, and a urine culture was collected upon admission, which yielded *E. faecalis*, which was considered as a superinfection. Twelve days after tocilizumab administration, *Klebsiella pneumoniae* was isolated in the blood cultures. The patient had a favorable outcome and was discharged from the hospital.

Patient 3 was a 70-year-old woman. Charlson index: five. SEIMC Score: eight. Prophylaxis with ceftriaxone was maintained for 7 days. They were admitted to the intensive care unit the following day. Five days after tocilizumab administration, *C. albicans* was isolated in a urine culture, which was treated with fluconazole, and *ESBL E. coli* was isolated in a rectal swab, which was considered as colonization. The patient had a favorable outcome and was discharged from the hospital.

Patient 4 was a woman of 50 years old. Charlson index: one. SEIMC Score: four. They used prophylaxis with ceftriaxone for 6 days. They were admitted to the intensive care unit on the same day as the administration of tocilizumab. On the 2nd day of ICU admission, *C. albicans* was isolated in a tracheal aspirate, leading to the initiation of fluconazole treatment, which was later switched to anidulafungin. On the 7th day of ICU admission, in the context of a fever and clinical deterioration, several microbiological samples were collected, intravenous ceftriaxone was discontinued, and broad-spectrum antibiotic therapy was expanded, but there was no clinical improvement. The patient passed away before the growth result of *E. faecalis* in the urine culture was known.

Patient 5 was a 73-year-old male. Charlson index: three. SEIMC Score: six. Prophylaxis with ceftriaxone was taken for 7 days. They were admitted to the intensive care unit on the fourth day after the administration of tocilizumab. On the sixth day after tocilizumab administration, *MRSA* was isolated from the nasal swab, which was interpreted as colonization. The patient did not survive intensive care unit admission.

Patient 6 was a 45-year-old woman. Charlson index: 0. SEIMC Score: three. Prophylaxis with ceftobiprole was maintained for 6 days. They were admitted to the intensive care unit on the day following the administration of tocilizumab. On the 6th day after tocilizumab administration, *C. albicans* was isolated in a urine culture, which was interpreted as a superinfection and treated with intravenous anidulafungin. The subsequent progress was satisfactory, and she was able to be discharged from the hospital after an extended admission.

Patient 7 was a 42-year-old female. Charlson index: two. SEIMC Score: five. Prophylaxis with the continuous infusion of ceftobiprole was taken for one day. They were admitted to the intensive care unit four days after tocilizumab administration. On the 7th day after tocilizumab administration, *P. aeruginosa* was isolated from the bronchial aspirate sample. This was interpreted as colonization. The patient passed away in the intensive care unit.

Patient 8 was a 62-year-old male. Charlson index: two. SEIMC Score: nine. Prophylaxis with meropenem plus linezolid were taken for 3 days. They were admitted to the intensive care unit two days after tocilizumab administration. On the 14th day after tocilizumab administration, *MRSA* was isolated from a nasal swab. This was interpreted as colonization. The patient survived the hospital admission and was discharged.

Patient 9 was a 70-year-old male. Charlson index: four. SEIMC Score: nine. Prophylaxis with meropenem plus linezolid were taken for 12 days. Tocilizumab was administered on the third day of ICU admission. On the third day after tocilizumab administration, *P. aeruginosa* was isolated from rectal exudate in the context of clinical symptoms, including a fever, elevated inflammatory markers, and diarrhea, which was considered a superinfection. On the 14th day, an *ESBL E.coli* was isolated from rectal exudate, which was interpreted as colonization. The patient passed away in the intensive care unit.

Patient 10 was a 60-year-old male. Charlson index: three. SEIMC Score: six. Prophylaxis with piperacillin plus tazobactam was started on the day of tocilizumab administration for one day. The patient was admitted to the intensive care unit the day after administration. On the 4th day, in the context of suspected respiratory superinfection, the IgM serology results for *Mycoplasma pneumoniae* came back positive, which was interpreted as superinfection. On the 6th day after tocilizumab administration, *Aspergillus fumigatus* was isolated in a bronchial aspirate sample, which was interpreted as a superinfection. The patient passed away in the intensive care unit.

Patient 11 was a 70-year-old woman. Charlson index: three. SEIMC Score: 10. She did not receive antibiotic prophylaxis. The patient was admitted to the intensive care unit on the fifth day after tocilizumab administration. On the day following admission to the ICU, *Enterobacter sakazakii* was isolated from a tracheal aspirate, which was interpreted as a superinfection, and intravenous antibiotic therapy was initiated. Eight days after admission to the ICU, *C. albicans* was isolated in a urine culture, and fluconazole was added to the treatment. On the 14th day following tocilizumab administration, *C. albicans* was isolated again from a tracheal aspirate, leading to a change in treatment to anidulafungin and voriconazole. The patient subsequently passed away in the ICU.

Patient 12 was a 77-year-old female. Charlson index: three. SEIMC Score: 14. She did not receive prior or simultaneous antibiotic prophylaxis with tocilizumab administration. On the 13th day after tocilizumab administration, *S. epidermidis* was isolated in blood cultures, and *E. faecalis* was isolated in a urine culture, with both isolations interpreted as superinfections. The patient passed away during hospitalization.

Patient 13 was a 55-year-old male. Charlson index: one. SEIMC Score: six. He did not receive prior or simultaneous antibiotic prophylaxis with tocilizumab administration. He was admitted to the intensive care unit on the same day as tocilizumab administration. On the 13th day after tocilizumab administration, *P. aeruginosa* and *C. albicans* were isolated from a bronchial aspirate sample, which was interpreted as colonization. The patient survived hospitalization and was discharged.

Patient 14 was an 82-year-old male. Charlson index: five. SEIMC Score: 20. He did not receive prior or simultaneous antibiotic prophylaxis with tocilizumab administration. On the 7th day after tocilizumab administration, *S. aureus* was isolated in the blood cultures, and he admitted to the intensive care unit. The patient passed away in the intensive care unit

## 4. Discussion

In this study, we describe the incidence and microbiological characteristics of bacterial and fungal colonizations/infections identified in patients hospitalized for COVID-19 and treated with steroids, following tocilizumab treatment, based on the use of antibiotic prophylaxis. In our cohort, 60.3% of the patients received antibiotic prophylaxis, mainly ceftriaxone and ceftobiprole. There were no significant clinical differences between the patients based on the use of prophylaxis, with a trend towards a higher number of comorbidities in the subgroup that received antibiotic prophylaxis. There were no significant differences between the two patient subgroups regarding the number of clinically diagnosed infections. However, in almost half of the patients who received antibiotic prophylaxis, colonization by a multidrug-resistant pathogen was identified, which was compared to no isolation with the growth of a multidrug-resistant pathogen in the subgroup that did not receive antibiotic prophylaxis.

This work, to our knowledge, represents the first published study on the effects of prior antibiotic therapy before the administration of the combination of tocilizumab and steroids in COVID-19 patients, aiming to prevent the development of clinical infections or promote colonization. One of the likely reasons for the lack of literature on this topic is the difficulty in obtaining a significant number of patients, as evidenced in this study, where only 3.4% of the patients admitted during the study period met the inclusion criteria. A total of 64.3% of the included patients received prophylaxis, with 11 different antibiotics, some of them in variable combinations, making it difficult to draw statistically significant differences. However, precisely because of the difficulty and the lack of published data on this clinical question that arises in the daily practice of infectious disease units and intensive care units worldwide, the analysis of this cohort is relevant for two reasons. First, to attempt to draw conclusions that help discern the decision to prescribing antibiotic therapy or not in these patients. Second, this study aims to serve as a basis and stimulus for scientific societies and international study groups to analyze existing multicenter cohorts of COVID-19 patients in the search of a larger number of patients to identify statistically significant trends.

The authors are aware that these data correspond to COVID-19 patients diagnosed in Spain in 2020 and that there could be changes in these results following the emergence of subsequent variants of COVID-19 with different clinical courses. Similarly, the effects of vaccination could impact these results. Nevertheless, the current number of patients with moderate or severe COVID-19 eligible for tocilizumab treatment is very low, which explains the absence of other studies related to this topic.

The possible biases and confounding factors are also presented. Bacterial superinfections might be underdiagnosed for several reasons: there were difficulties in sample collection due to overwhelming hospital services during the first wave of the pandemic [16], there are a high number of bacterial pneumonia cases where microbiological diagnosis is not achieved, as is well known in routine clinical practice [17,18], and PCR diagnostic techniques are not used for bacterial identification in the analyzed samples. Another potential limitation of this study, given its observational nature, is that the risk of developing an infection might have been higher in the group of patients who received prophylaxis, and therefore, despite finding similar results between the two patient subgroups, this could be interpreted as a protective role of antibiotic prophylaxis. Interestingly, although not reaching statistically significant differences, the group of patients who received antibiotic prophylaxis showed a significant trend towards having more pre-existing comorbidities before the onset of the clinical condition [14,19]. Since the median comorbidity score was higher for the group who took antibiotic prophylaxis, without propensity scoring, the comparison between the cases and control may not be accurate. It could be that this group is sicker and that it, in the past, they have needed more antibiotics, which leads to development of MDR bacteria [5].

A total of 14% of the total included patients developed a bacterial and/or fungal superinfection, without observing differences based on whether they had received antibiotic prophylaxis or not. In the retrospective cohort of hospitalized patients with COVID-19 published by Moreno et al. [20], where 82 out of the 306 patients included were treated with tocilizumab, 14% of the patients experienced a severe infection, and 26% of those were admitted to the ICU. However, in this cohort, there was no differentiation in the incidence of infections based on whether tocilizumab had been administered. In this study, it is not indicated whether the patients received antibiotic prophylaxis. However, 76.6% of the patients treated with tocilizumab received azithromycin for the treatment of COVID pneumonia.

In our study, 40% of the patients who received prophylaxis were treated with ceftriaxone, and 31.3% were treated with ceftobiprole, allowing some consequential case descriptions. This is particularly important as these are two antimicrobials with different, and in the case of ceftobiprole, broader spectra of action. The patients who received prophylaxis with ceftriaxone had a higher incidence of superinfections (22.2%) compared to those of the subgroup of patients who received prophylaxis with ceftobiprole (7.1%) and the subgroup that did not receive prophylaxis (12%). We could infer that an ineffective empirical antibiotic therapy, without the coverage of some bacteria that most frequently superinfect immunosuppressed COVID-19 patients (*MRSA*, *Pseudomonas*, *E. faecalis*) [4,21,22,23] could alter the patients’ commensal flora, thereby creating an ecological niche for these pathogens to subsequently infect patients more easily than would have occurred without unnecessary prior antibiotic therapy. This would explain cases like that of patient 1, with up to four isolations, where three of them could be interpreted as colonization, detected within 14 days after tocilizumab infusion. In fact, in this study, we did not find any superinfections caused by *S. pneumoniae*, despite it being another of the most frequently identified bacteria causing superinfections in COVID-19 [4,21,22,23]. When comparing the most frequently bacterial isolates in our patient cohort, a urinary infection due to *E. faecalis* stands out in three patients, two of whom received ceftriaxone as a prophylaxis. In the García-Vidal et al. [4] cohort, there were only two cases of a urinary superinfection caused by *E. faecalis* out of the 74 documented bacterial infections; it is possible that the use of ceftriaxone may have contributed to the higher incidence of this microorganism in our study.

In reviewing the patients who received other antibiotic therapies, the isolations detected in patients treated with the combination of meropenem and linezolid (patients 8 and 9) are striking. Despite these patients receiving prophylaxis that, in theory, should have eliminated non-multidrug-resistant *P. aeruginosa,* as in the cases of meropenem [24,25] and MRSA, including linezolid [26], colonization by MRSA (patient 8) and a superinfection caused by *P. aeruginosa* were detected. In the patients treated with ceftobiprole, no isolates of bacteria included in its microbiological coverage were identified. In the group of patients who received prophylaxis with ceftriaxone, only one patient presented a microbiological isolate that was initially covered by the bacterial spectrum of the chosen antibiotic prophylaxis (patient 4, bacteremia caused by ceftriaxone-sensitive *K. pneumoniae*).

In our study, the percentage of superinfections caused by multi-drug-resistant bacteria is similar to what has been reported in other series [4,27,28]. However, it is noteworthy that there were no occurrences of colonization/superinfections by carbapenem-resistant Gram-negative bacteria, nor the isolation of *Acinetobacter* spp. This is in contrast to the increased prevalence of these during the COVID-19 epidemic, as reported in several studies [27,28]. Mady et al. [29] evaluated the use of tocilizumab in patients with severe COVID-19 pneumonia requiring ICU admission. All the patients included in this series were also given intravenous dexamethasone and prophylactic antibiotics (azithromycin, meropenem, vancomycin, or piperacillin/tazobactam for 14 days). Twelve out of the sixty-one patients included (19%) developed a superinfection. The most frequent microorganisms were *Acinetobacter baumanii* and *Pseudomonas* spp. The local ecology of our hospital and limited use of meropenem as a prophylaxis might account for our findings. On the other hand, our study highlights the isolation of *ESBL E. coli* in the rectal swabs from three patients. This finding contrasts with the results of a systematic review published by Abubakar et al. [30]. In this meta-analysis, they found a reduction in the prevalence of *ESBL E. coli* infections during the pandemic, which could be attributed to infection control strategies. In our case, the predominant use of cephalosporins as an antibiotic prophylaxis could explain the increased prevalence of this pathogen.

Regarding fungal infections, in our study, it is notable that *C. albicans* was the most frequently isolated microorganism, which was found in six patients. One patient had an isolation of *C. albicans* in urine and another isolation in a tracheal aspirate sample. All the patients with the *C. albicans* isolation were admitted to the ICU after the administration of tocilizumab. In three patients, it was identified in respiratory tract samples, and in three patients, it was detected in urine cultures (Table 2). The interpretation of the colonization or pathogenic role of *C. albicans* isolates in respiratory samples from critically ill patients is highly debated, given the challenge of obtaining lung biopsies to confirm the presence of *C. albicans* in pulmonary parenchyma associated with inflammation. In general, the isolation of *C. albicans* in lower respiratory tract samples is interpreted as colonization, considering the rarity of pulmonary candidiasis [31]. In all the patients in whom *C. albicans* was isolated in both respiratory and urine cultures, the clinicians made the decision to add an antifungal treatment. However, the authors of this article have chosen to classify isolations from respiratory samples as colonizations. Nevertheless, this decision may be subject to questioning. The treatment with tocilizumab combined with systemic steroids has been shown to be a risk factor for the acquisition of *Candida* spp. in COVID-19 patients [32]. Experimental studies performed on mice deficient in interleukin-6 have demonstrated the role of this interleukin in the defense against *Candida* infections [33,34]. The microbiological results of our cohort of patients treated with tocilizumab support these experimental studies. It would be of great interest to conduct clinical studies to assess whether antifungal prophylaxis prior to tocilizumab treatment could prevent these *Candida* colonizations/infections.

In conclusion, in our cohort of patients hospitalized for COVID-19, treated with steroids, and who received tocilizumab, the use of antibiotic prophylaxis did not reduce the incidence of secondary bacterial infections. However, after the administration of prophylaxis, the patients showed an increase in the incidence of colonization by multidrug-resistant bacteria.

## Figures and Tables

**Figure 1 antibiotics-12-01515-f001:**
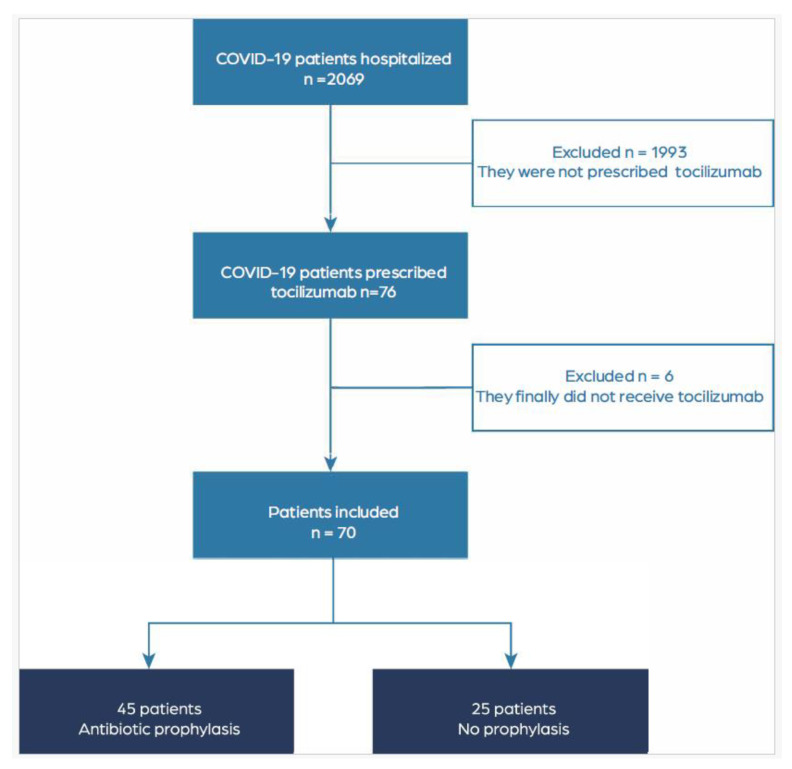
Flowchart of patient selection for the study.

**Table 1 antibiotics-12-01515-t001:** Characteristics of study population.

Characteristic	All (n = 70)	Profilaxis (n = 45)	No Profilaxis (n = 25)	*p* Value
Age (years), median (IQR)	66 (54–77)	66 (54–77)	65 (55–78)	0.75
Charlson, median (IQR)	3 (2–5)	3 (2–5)	2 (1–4)	0.09
SEIMC score, median (IQR)	10 (5–15)	10 (5–15)	9 (5–13)	0.56
All isolates (bacterial/fungical), n	25 (18/7)	18 (13/5)	7 (5/2)	0.32
Patients with microbiological isolates, n (%)	14 (20)	10 (22.2)	4 (16)	0.53
Bacterial/fungical superinfections, n	14 (10/4)	9 (6/3)	5 (4/1)	1
Patients with bacterial/fungical superinfection, n (%)	10 (14.2)	7 (15.5)	3 (12)	0.68
Focus of superinfection, n				
Lung	4	3	1	
Urinary	6	4	2	
Bloodstream	3	1	2	
Rectal	1	1	0	
ICU admission, n (%)	20 (28.6)	11 (24.4)	9 (36)	0.30
Hospital mortality, N (%)	32 (45.7)	19 (42.2)	13 (52)	0.43

IQR: interquartile range. ICU: intensive care unit. *p* values: Chi square test.

**Table 2 antibiotics-12-01515-t002:** Description of microbiological isolates.

Patient	Prophylaxis	Microbiological Isolate	Source of Infection	Days since Tocilizumab Administration	Outcome	Superinfection
1	Ceftriaxone	*S. epidermidis*	bloodstream	8	Death	No
1	Ceftriaxone	*C. albicans*	respiratory	8	Death	No
1	Ceftriaxone	*Chlamydia pneumoniae*	respiratory	8	Death	Yes
1	Ceftriaxone	*ESBL E. coli*	rectal	12	Death	No
2	Ceftriaxone	*E. faecalis*	urinary	1	Discharge	Yes
2	Ceftriaxone	*K. pneumoniae*	bloodstream	12	Discharge	Yes
3	Ceftriaxone	*C. albicans*	urinary	5	Discharge	Yes
3	Ceftriaxone	*ESBL E. coli*	rectal	5	Discharge	No
4	Ceftriaxone	*C. albicans*	respiratory	2	Death	No
4	Ceftriaxone	*E. faecalis*	urinary	7	Death	Yes
5	Ceftriaxone	*S. aureus*	nasal	6	Death	No
6	Ceftobiprole	*C. albicans*	urinary	6	Discharge	Yes
7	Ceftobiprole	*P. aeruginosa*	respiratory	7	Death	No
8	Meropenem plus linezolid	*MRSA*	nasal	14	Discharge	No
9	Meropenem plus linezolid	*P. aeruginosa*	rectal	3	Death	Yes
9	Meropenem plus linezolid	*ESBL E. coIi*	rectal	14	Death	No
10	Piperacillin - tazobactam	*Mycoplasma pneumoniae*	respiratory	4	Death	Yes
10	Piperacillin - tazobactam plus teicoplanin	*A. fumigatus*	respiratory	6	Death	Yes
11	None	*Enterobacter* spp.	respiratory	6	Death	Yes
11	None	*C. albicans*	urinary	13	Death	Yes
11	None	*C.albicans*	respiratory	14	Death	No
12	None	*S. epidermidis*	bloodstream	13	Death	Yes
12	None	*E. faecalis*	urinary	13	Death	Yes
13	None	*P. aeruginosa*	respiratory	13	Discharge	No
13	None	*C.albicans*	respiratory	13	Discharge	No
14	None	*S.aureus*	bloodstream	7	Death	Yes

MRSA: methicillin-resistant Staphylococcus aureus. ESBL: extended-spectrum β-lactamases.

**Table 3 antibiotics-12-01515-t003:** Outcomes according to antimicrobial prophylaxis prior to tocilizumab.

Antibiotic	N	Bacterial/Fungical Isolates, n	Patients with Microbiological Isolates, n (%)	Patients with Superinfection, n (%)	Microbiological Isolates as Colonization (Bacterial/Fungical)	Hospital Mortality, n (%)
Ceftriaxone	18	11 (8/3)	5 (27.8%)	4 (22.2)	6 (4/2)	6 (33.3%)
Ceftobiprole	14	2 (1/1)	2 (14.2%)	1 (7.1)	1 (1/0)	7 (50%)
Other	13	5 (4/1)	3 (23%)	2 (15.4)	2 (2/0)	6 (46%)
No prophylaxis	25	8 (5/3)	4 (16%)	3 (12)	3 (1/2)	13 (52%)
All	70	26 (18/8)	14 (20%)	10 (14.2)	12 (8/4)	32 (45.7%)

## Data Availability

Data are not public due to restrictions by Ministry of Defence of the Kingdom of Spain. These results were partially presented at IdWeek 2021.

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
