# Peer review of "The Impact of Antibiotic Prophylaxis on a Retrospective Cohort of Hospitalized Patients with COVID-19 Treated with a Combination of Steroids and Tocilizumab"

_antibiotics, 2023, doi:10.3390/antibiotics12101515_

Round 1

Reviewer 1 Report

The issue of Antimicrobial resistance is important and this paper throws light on the same.

Is the SEIMC score an acronym? If the authors can describe in a couple of lines, what are the components, it will be clear for the reader. The components that make up Charlson's score could be mentioned as well.

The study process is very well defined. A flowchart displaying the numbers would be great and give a good snapshot to the reader.

I understand given the small sample size, advanced statistical analyses will not be possible. Please provide the p-values in Table 1. Since the median comorbidity score was higher for the group with the antibiotic prophylaxis, without propensity scoring the comparison between cases and control may not be accurate.  It could be that this group is sicker and that it in the past they have needed more antibiotics which leads to development of AMR.

Can the AWARE classification be used to classify the antibiotics? If this classification is used, I assume Table 2 will get simpler and Fisher's exact could be calculated. Give it a try.

Author Response

The issue of Antimicrobial resistance is important and this paper throws light on the same.

Is the SEIMC score an acronym? If the authors can describe in a couple of lines, what are the components, it will be clear for the reader. The components that make up Charlson's score could be mentioned as well.

R: Thank you for your comment. SEIMC is the acronym of the group that has elaborated this score (Sociedad Española de Enfermedades Infecciosas y Microbiología Clínica, in english Spanish Society of Infectious Diseases and Clinical Microbiology). We have added it in the manuscript. SEIMC score is described in ref. 15. We can also mention in the manuscript the components of Charlson’s score.

The study process is very well defined. A flowchart displaying the numbers would be great and give a good snapshot to the reader.

R: I agreed with the reviewer that is necessary a flowchart. We have added a flowchart as figure 1.

I understand given the small sample size, advanced statistical analyses will not be possible. Please provide the p-values in Table 1.

R: Thank you for your comment. We have added it.

Since the median comorbidity score was higher for the group with the antibiotic prophylaxis, without propensity scoring the comparison between cases and control may not be accurate.  It could be that this group is sicker and that it in the past they have needed more antibiotics which leads to development of AMR.

R: We agree with the reviewer, and we have added a comment in the discussion section, explaining this limitation of the study.

Can the AWARE classification be used to classify the antibiotics? If this classification is used, I assume Table 2 will get simpler and Fisher's exact could be calculated. Give it a try.

R: The proposal to classify antibiotics used in prophylaxis according to the AWaRe classification is very appropriate; however, we were unable to do so because there are several patients receiving combinations of beta-lactams and linezolid, in which the beta-lactam (such as piperacillin/tazobactam or meropenem) would be classified as Watch, and linezolid as Reserve; thus, we cannot determine which group of antibiotics these combinations would belong to.

Reviewer 2 Report

Dear authors,

Thank you for the study, I almost reject the manuscript outright. But after an extensive consideration I came to the following recommendations. Please back up these claims with a most recent citations.

1. `The immunosuppressive treatment used for managing COVID-19 has been considered a potential risk factor for developing nosocomial acquired infections'

2Investigation of bacterial and fungal pathogens in blood, normally sterile fluids, sputum and other samples was performed with standard microbiologic procedures during hospitalization, as requested by the attending physician.

3. Microbiological isolates within 14 days following the administration of tocilizumab were considered. (perhaps: Fractional derivative operator on quarantine and isolation principle for COVID-19, might be of a suitable citation).

4. What do you mean with `our hospital'?

5. Since the data in this study, is for Spain 2020, what is the significant of this research work in terms of the development that has taken places in Spain beyond 2020 with regard to treating COVID-19 patients? Please clarify this with extensive analysis and most recent citations.

6. Interested readers, would want to know how is your work different form `

Impact of Antibiotic Prophylaxis Prior to Treatment with Steroids and Tocilizumab in COVID-19 Patients'? and why this work is not cited in this manuscript?

Read prove the entire manuscript for typos.

Unless the above recommendations are attended to, otherwise I do not recommend this manuscript for publication.

Author Response

  1. `The immunosuppressive treatment used for managing COVID-19 has been considered a potential risk factor for developing nosocomial acquired infections'

R: Reference number 5 is added

  1. Investigation of bacterial and fungal pathogens in blood, normally sterile fluids, sputum and other samples was performed with standard microbiologic procedures during hospitalization, as requested by the attending physician.

R: An additional paragraph is added in “Procedures” detailing the references that were followed for the processing of the samples.

  1. Microbiological isolates within 14 days following the administration of tocilizumab were considered. (perhaps: Fractional derivative operator on quarantine and isolation principle for COVID-19, might be of a suitable citation).

R: Reference number 13 is added.

  1. What do you mean with `our hospital'?

R: Hospital Central de la Defensa “Gómez Ulla”, the hospital where we have developed the study. It is described in the first reference to the hospital, in “Study design and patients”, paragraph 1, line 1.

  1. Since the data in this study, is for Spain 2020, what is the significant of this research work in terms of the development that has taken places in Spain beyond 2020 with regard to treating COVID-19 patients? Please clarify this with extensive analysis and most recent citations.

R: This issue is described in “discussion”, paragraph 2. Anyway, we have added new analysis in a new paragraph, now “discussion”, paragraph 3.

  1. Interested readers, would want to know how is your work different form `Impact of Antibiotic Prophylaxis Prior to Treatment with Steroids and Tocilizumab in COVID-19 Patients'? and why this work is not cited in this manuscript?

R: This reference is related with a partial presentation of our results in IdWeek 2021. We have added a disclosure informing this issue.

Read prove the entire manuscript for typos.

R: We have reviewed the manuscript and corrected typographic errors.

Reviewer 3 Report

A well written paper that requires some minor tidying up:

Abstract line 14: Replace the word “ germs” with something more scientific e.g. bacteria or pathogens…… Check throughout the paper – e.g. page 4.

Page 3: Procedures:

I would avoid the use of “CPE” as an abbreviation for carbapenem-resistant Pseudomonas aeruginosa as “CPE” is widely used in the literature as an abbreviation for carbapenemase-producing Enterobacterales. If an abbreviation is justified, I suggests using “CRPA”.

Ensure the use of italics for names of microbes. Check these points throughout the whole paper. There is currently inconsistent use of italics e.g. For C. albicans (page 4).

Page 4: The authors state “24.4% of patients who required antibiotic prophylaxis….” I would change the word ‘required’ to ‘received’. Whether they “required” prophylaxis or not would appear to be part of what is being investigated by this study.

Page 4: Please ensure that when bacterial names are first mentioned they are provided in full e.g. Escherichia coli rather than E. coli. Apply this approach throughout the whole paper for other microbes.

Only very minor editing of language is required.

Author Response

A well written paper that requires some minor tidying up:

R: Thank you for your comment.

Abstract line 14: Replace the word “ germs” with something more scientific e.g. bacteria or pathogens…… Check throughout the paper – e.g. page 4.

R: Thank you. We have changed it.

Page 3: Procedures:

I would avoid the use of “CPE” as an abbreviation for carbapenem-resistant Pseudomonas aeruginosa as “CPE” is widely used in the literature as an abbreviation for carbapenemase-producing Enterobacterales. If an abbreviation is justified, I suggests using “CRPA”.

R: Thank you. We have changed it in the manuscript.

Ensure the use of italics for names of microbes. Check these points throughout the whole paper. There is currently inconsistent use of italics e.g. For C. albicans (page 4).

R: Thank you for your comment. We have reviewed the manuscript and changed it.

Page 4: The authors state “24.4% of patients who required antibiotic prophylaxis….” I would change the word ‘required’ to ‘received’. Whether they “required” prophylaxis or not would appear to be part of what is being investigated by this study.

R: We agree with the reviewer. We have changed it in the manuscript.

Page 4: Please ensure that when bacterial names are first mentioned they are provided in full e.g. Escherichia coli rather than E. coli. Apply this approach throughout the whole paper for other microbes.

R: Thank you for your suggestion. We have reviewed the manuscript and change it.

Round 2

Reviewer 1 Report

Thanks for giving me an opportunity to review your study. You have incorporated all the suggestions I made.

- The flowchart makes the study very lucid to the reader.

- Table 1: Please indicate which test whose p-value is indicated. Fisher's for categorical and Mann Whitney for continuous ones. The word prophylaxis is misspelt. 

Author Response

Fortunately you made this comment. We discovered a mistake done in previous versions of the manuscript. Statistical test used for table 1 was chi square test. It has been corrected both in the test and in table 1.

Sorry for the mistake with the spelling of "prophylaxis". It has been corrected.

Regards

Reviewer 2 Report

Dear author,

Thank you for attending to the recommendations.

Author Response

Thank you too for your suggestions, which have enrichened the manuscript.